# Limb–Girdle Muscular Dystrophy D2 TNPO3-Related: A Quality of Life Study

Alicia Aurora Rodríguez [1],*, Imanol Amayra [1], Irune García [1] and Corrado Angelini [2]

1   Neuro-e-Motion Research Team, Faculty of Health Sciences, University of Deusto, Av. Universidades, 24, 48007 Bilbao, Spain; imanol.amayra@deusto.es (I.A.); irune.garciaurquiza@deusto.es (I.G.)
2   Neuromuscular Laboratory, Department of Neurosciences, University of Padova, Campus Biomedico Pietro d'Abano, 35131 Padua, Italy; corrado.angelini@unipd.it
*   Correspondence: aliciarodriguez.b@deusto.es; Tel.: +34-944-139-000 (ext. 2577)

**Abstract:** The present study is the first research that analyzes the quality of life (QoL) of people affected by a dominant form of limb–girdle muscular dystrophy, specifically limb–girdle muscular dystrophy D2 (LGMD-D2). Additionally, clinical forms of the individual cases of the six affected patients are presented. This study also aims to explore the differences between patients' reports and caregivers' reports, and between LGMD-D2 and recessive forms of LGMD. The instruments used were as follows: sociodemographic data, GSGC, and INQoL instrument. The sample consisted of six people affected by LGMD-D2: three caregivers of three affected people, and three patients with recessive LGMD. They came from associations of affected people and a hospital in Padua. Those affected have multiple symptoms that lead to disability, which ultimately leads to dependence on the assistance. The present study shows that LGMD-D2 has a greater impact on activities of daily living, fatigue, muscle pain, and independence than other LGMD pathologies or other neuromuscular diseases. It also appears that age could influence QoL, and that muscle weakness is a very disabling symptom in this variant. In the current context of constantly developing research for new treatments, it is essential to analyze which aspects are most affected. Finally, caregivers can play an essential role in symptom reporting, as certain psychological adjustment mechanisms in the patient may be interfering with the objectivity of the report.

**Keywords:** LGMD-D2; quality of life; caregivers





## 1. Introduction

Limb–girdle muscular dystrophy (LGMD) is a group of genetic conditions characterized by progressive muscle weakness and the loss of muscle mass, predominantly affecting the proximal muscles of the limbs [1,2]. They are part of the diagnostic group of neuromuscular diseases (NMDs) and are classified into autosomal dominant and autosomal recessive conditions [3]. About 10% of these dystrophies are dominant and 90% are recessive [1]. Currently, more than 30 distinct genetic subtypes of LGMD have been identified [4] and are grouped according to inheritance patterns. Severe childhood-onset forms, distal and proximal myopathies, pseudo-metabolic myopathies, eosinophilic myositis, and a higher average serum creatine kinase (CK) are some of the common clinical phenotypes that usually occur due to mutation in LGMD genes [5]. Following a revision of their classification by the scientific community, autosomal dominant LGMDs are referred to as D and numbered 1 to 5, and recessive forms as R with numbers 1 to 23. Despite the enormous genetic heterogeneity and the increasingly rapid identification of new disease genes, nomenclature has become a major problem as not all have an associated nomenclature [6]. They have an overall prevalence of approximately four to seven per a hundred thousand people, depending on the region [6]. In general, recessive forms are clinically more severe than dominant forms. Dominant forms have a lower average CK than recessive forms [7] and disease progression is generally slower [7,8].

The onset of the disease can range from childhood to adulthood with progressive muscle weakness. Specific muscles that are affected are the hip, pelvis, upper arm, and shoulder girdle, making gait difficult [9,10].

Research studies on these pathologies are limited, and when involving low-prevalence subtypes of LGMD, this issue is even more pronounced. This is the case of the dominant form of LGMD-D2. In this case, due to the association of this disease with the TNPO3 gene, the protein "transportin-3" is affected and is therefore called "transportinopathy" or "LGMD D2 TNPO3-related". This variant produces both early and late-onset phenotypes and was identified in familial and sporadic cases [1]. It was first described in an Italo-Spanish family with proximal axial and girdle muscle weakness [11], and its prevalence is currently unknown, with 32 individuals identified in 2001 in a large Spanish kindred spanning five generations [12]. The family was found to have a high variability in terms of age and symptom onset, ranging from 1 to 31 years, and in the degree of involvement of the pelvic girdle muscles. However, one characteristic they all had in common was a generalized atrophy of muscle mass [11]. These symptoms can lead to disability, producing dependence on the community and close relatives. DNA genetic research identified a heterozygous mutation in the determination codon of the TNPO3 gene [13,14]. TNPO3 encodes transportin-3, which is part of the beta-caryopherin family of nuclear import/export receptors. TNPO3 is an importin, which binds and transports proteins containing serine/arginine (SR) domains from the cytoplasm to the nucleus [8].

This autosomal dominant LGMD was mapped to 7q32.1–32.2. Then, a single nucleotide deletion resulting in a frameshift (NM_012470.3:c.2771delA) in TNPO3 gene that extends the protein by 15 amino acids was found to be responsible by next-generation sequencing. On the other hand, a Hungarian family, another family identified with LGMD-D2 symptomatology, was described with early onset LGMD caused by a heterozygous frameshift variant (c.2767delC p. (Arg923AspfsTer17)) in TNPO3. Similar to the large Italo-Spanish family, this single pathogenic variant extends the C' end of the TNPO3 protein by 15 amino acids [8]. Although the different variants of the mutation may express similar phenotypes, it was found that in the Italo-Spanish family, the weakness usually appeared from the age of 15 years. However, in the Hungarian family, it appeared from the first year of life. Additionally, the pattern of weakness also differed to some extent, affecting the facial and bulbar levels in the case of the Hungarian family [8].

Currently, LGMD has no curative treatment [15,16]. All these treatments are symptomatic, and it is necessary to consider that the symptomatology produced by the disease can limit life expectancy and reduce quality of life (QoL) [17,18]. Thus, patients typically report limitations in aspects such as mobility, difficulty in performing activities, limitations in social interaction, as well as emotional impact [17]. Furthermore, to generate or create new treatments aimed at improving the lives of these people, it is especially important to know which aspects of their QoL are most affected, so that the treatment can be as specific as possible [19]. The World Health Organization (WHO) defines QoL as a faceted concept, reflecting the physical, psychological, and social conditions of the person [20–22].

In chronic diseases and NMD, there is a decline in the QoL experienced by the patient [23–25]. These studies, which analyze the QoL in LGMD, have mostly been in other recessive forms. Therefore, they are limited in this group and, if we focus on the subtypes of LGMD, they are often non-existent.

The analysis of the QoL of people with muscular dystrophies has allowed a better understanding of how patients experience this disease, because, as discussed above, most of them currently have no effective medical treatment. On the other hand, the analysis of family members' reports on what aspects are most affected in the lives of affected people helps us to gain another perspective and to learn about the disease from different sources. These studies can identify areas to target interventions that could help to preserve the QoL despite disease progression. In addition, they also serve to examine the efficacy of the interventions undertaken [26]. Therefore, it is also important to analyze whether a low QoL is also related to one autosomal form or the other.

Therefore, the present study aims to present the clinical forms of six patients in more depth. In addition, the aim is to explore for the first time the QoL in patients with LGMD-D2, from the point of view of the patient and the caregiver, and comparing it to a sample of patients with recessive forms of LGMD.

## 2. Methods

### 2.1. Participants

The sample was composed of 6 people affected by LGMD-D2 and 3 caregivers of 3 affected people. They came from associations of patients and from a hospital in Padua. Data were also collected from 3 patients affected by recessive forms of LGMD.

The inclusion and exclusion criteria were as follows:

Inclusion criteria: (a) the presence of diagnosis of LGMD or being an informal caregiver of a person diagnosed with LGMD; (b) signing the informed consent document before participating in the study; (c) one of their main languages of communication being Spanish or Italian.

Exclusion criteria: (a) suffering from any other diagnosis not related to the diagnosis of LGMD; (b) suffering from any other psychological or psychiatric diagnosis not related to LGMD; (c) having uncompensated sensory deficits that preclude the administration of the assessment protocol; and (d) illiteracy.

### 2.2. Procedure

Recruitment of the sample was conducted through an association of patients and at a hospital in Padua. Those interested in joining the study were informed of the assessment procedure. Data collection was carried out via videoconference, but also in person, through a hybrid model. The participants signed the informed consent form before filling out the questionnaires, thus consenting to participate in the research. They also signed their informed consent for the use of the tests, including muscle MRI. The duration of the protocol was approximately half an hour. The study was approved by the Commission for Responsible Ethics (Ref: ETK-39/18-19); for the human material, laboratory studies, and muscle biopsy, consent was granted by the Ethical Committee of the University Hospital of Padua on 1 March 1995 and the investigations were conducted, following the rules of the 1975 Declaration of Helsinki.

### 2.3. Sociodemographic Data

Sociodemographic data were collected before the interview began, referring to the participants' gender and age.

### 2.4. Gait-Scale-Gowers-Chair (GSGC)

The GSGC score was used to evaluate functional performance. The GSGC test includes 4 items. The scale provides a holistic overview of the motor function of the patient being evaluated through a qualitative, timed measurement of four activities: G (gait) = walking for 10 m, S (scale) = climbing 4 steps of stairs, G (Gowers) = Gowers maneuver, C (chair) = rising from a chair. The final GSGC score is calculated by summing the scores of the four functional tests and varies from a minimum of 4 (normal performance) to a maximum of 27 (worst performance). The minimum score that can be obtained in each of the 4 categories is 1, and the maximum score is 7, except for the activity of rising from a chair, which is 6.

### 2.5. Quality of Life

The instrument used was the INQoL [27], Italian version [28]. INQoL reflects the physical and mental constraints resulting from the muscle condition.

It is an instrument that measures the QoL in people with NMD using 45 questions within 10 sections. Four sections refer to the impact of common muscle disease symptoms (weakness, locking, pain, and fatigue). Five sections examine the extent and significance

of the impact of muscle disorder on specific areas of life (activities, independence, relationships, emotions, and body image). These five sections refer to how the disease affects the emotional level, and how the person looks physically. In addition, it inquires how it influences their daily life activities, their independence, and their relationships with others. The total INQoL score is estimated from the five sections that measure the influence of muscle disease on specific areas of life. The last section is related to the treatment of the disease and was not used in our study. The final score for each subscale is presented as a percentage of the maximum negative impact. The closer the score is to 100%, the worse the aspects affects QoL [29].

### 2.6. Statistical Analysis

The statistical analysis was conducted using IBM SPSS Statistics for Windows (version 26). Demographic data were described using descriptive measures. Continuous variables were expressed as mean and standard deviation, and categorical variables as frequency and percentage. The Shapiro–Wilk test was used to check the normality of the variables. After determining the normal distribution, data were analyzed with Spearman's rank correlation coefficient. Statistical significance was inferred at $\alpha = 0.05$.

## 3. Results

### 3.1. Patients

Clinical data and classified disease progression were collected from six patients. Patients with early onset had a more severe phenotype, presenting a rapid disease course; however, patients with adult onset had a slow disease course. Muscle MRI showed a prominent atrophy of lower limb muscles, involving especially the vastus lateralis. Six detailed cases from two families affected by LGMD-D2 are presented below. These case reports are intended to present a more in-depth clinical perspective of the disease.

#### 3.1.1. The First Case of Family 1 (Patient 1)

*Patient 1*, an 18-year-old Hungarian child, weighs 28 kg, and his height is 150 cm. His GSGC score is 26 (G = 7, S = 7, G = 7, C = 6). Muscle MRI showed connective tissue infiltration in the lower limbs (Figure 1C,D) and at the time of the QoL interview, he was wheelchair-bound. Scores obtained using the Medical Research Council (MRC) scale are presented in Table 1. LGMD was caused by a heterozygous frameshift variant (c.2767delC P. (Arg923AspfsTer17)) in TNPO3. He showed weakness from the age of one year. He was able to walk at 18 months, but could not rise easily from the floor. At 10 years, he walked at a slow pace, up to about 3 km, but after that, he felt fatigued and fell down frequently. At 12 years, he was walking only 50–100 m because of his weakness. He could not rise from a lying position without gripping his knees and had a weak grip. He had problems swallowing with moderate dysphagia. CK was normal. He had no respiratory or cardiac involvement. He attends school by car and has no learning difficulties. In March 2018, he showed a waddling gait, Gowers' sign, and climbed stairs using the handrail. He had a myopathic face and moderate scapular wings. He was unable to stand on the floor or lift his legs off the bed. He appeared thin, frail, and weak. Deep tendon reflexes were absent.

#### 3.1.2. Second Case of Family 1 (Patient 2)

A 48-year-old woman of Hungarian origin, weighs 50 kg, and her height is 156 cm. She obtained a GSGC score of 22 (G = 4, S = 6, G = 6, C = 5). Muscle MRI showed posterior thigh muscles, i.e., semitendinosus and semimembranosus muscles, appearing less involved in the dystrophic process (Figure 1A). Scores obtained through the MRC scale are presented in Table 1. The LGMD was caused by a heterozygous frameshift variant (c.2767delC P. (Arg923AspfsTer17)) in TNPO3. She had no cardiac involvement or cognitive impairment. She has two sons, the first, *Patient 1*, is affected by LGMD-D2, but not the second.

The woman developed her first symptoms in childhood. She started walking at the age of 15 months and was always weak compared to her peers. At the age of six, she

underwent electromyography (EMG) and showed myopathic alterations. During her second pregnancy at the age of 30, she had a C-section and presented difficulty in breathing, feeling 'paralyzed'. At the age of 41, she had elevated CK, then presented with a decreased muscle strength in the forearm and finger extensors and in the back muscles. She also had proximal weakness involving mainly the proximal muscles of the lower extremities. At the last exam, at the age of 43, she was able to walk but had difficulty running. She reported frequent falls, had difficulty maintaining her legs elevated or off the ground (Gower's maneuver), and had to climb stairs using the handrail. She had a reduced forced vital capacity (FVC) on spirometry (by 57%) and had difficulty speaking and swallowing.

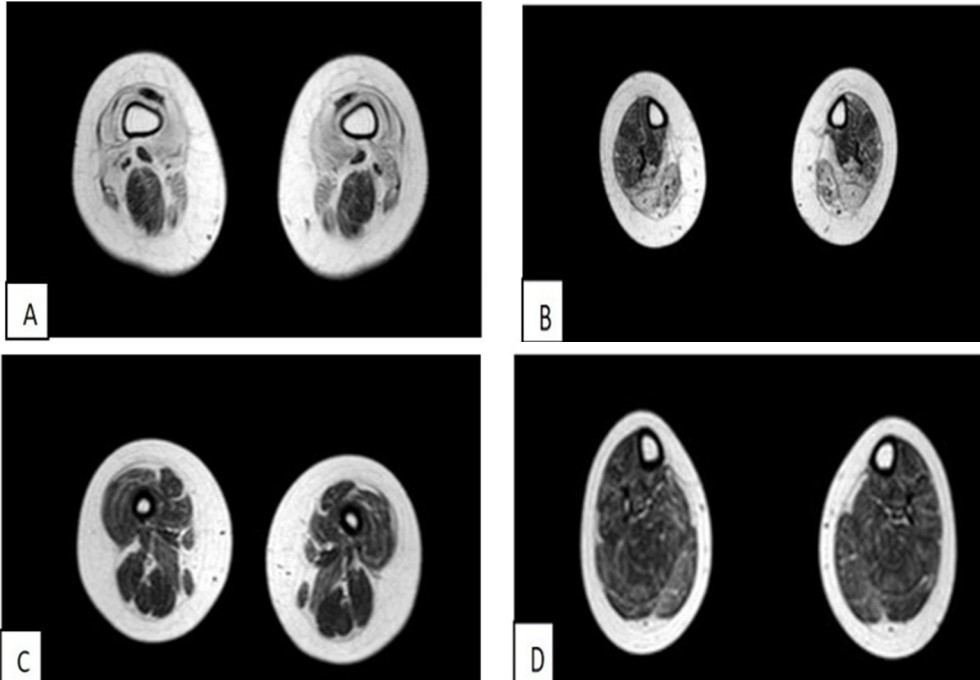

**Figure 1.** *Patient 1:* Muscles are characterized by hypotrophy in anterior thigh compartment (**C**). Gastrocnemius and soleus muscles were less compromised (**D**). *Patient 2*: Posterior thigh muscles, i.e., semitendinosus and semimembranosus muscles appear less involved in the dystrophic process (**A**). Muscles of posterior leg compartment are substituted with fatty infiltration (**B**).

**Table 1.** Scores obtained for each patient using MRC scale. Only scores for patient 1, 2, and 3 were collected.

| | Triceps | Biceps | Deltoid | Wrist Extensor | Wrist Flexor | Neck Flexor Muscles | Quadriceps | Iliopsoas | Semitendinosus Muscles | Semimembranosus Muscles | Tibialis Anterior Muscle |
|---|---|---|---|---|---|---|---|---|---|---|---|
| Patient 1 | 2/5 | - | 3/5 | 3/5 | - | 3/5 | 3/5 | 3/5 | 3/5 | 3/5 | - |
| Patient 2 | 2/5 | 4/5 | 2/5 | 3/5 | - | - | 3/5 | - | 4/5 | 4/5 | - |
| Patient 3 | 3/3 | 4-/4- | 3+/3+ | 4/4- | 4-/4- | - | 4-/4- | 3+/3- | - | - | 5/5 |

### 3.1.3. First Clinical Case of Family 2 (Patient 3)

This woman is 39 years old of Spanish origin, weighs 54 kg, and her height is 169 cm. She can stand and walk a few steps with aid, her grip strength is 4.8 kg (normal 27 kg), and a Dexa scan revealed femoral osteopenia. The DEXA scanning quantifies bone mineral density, indicating the likelihood of sustaining future fractures [30]. A muscle MRI showed atrophic thin muscles with advanced connective tissue substitution and EMG was normal. She obtained a GSGC score of 24 (gait = 7, scale = 7, Gowers = 7, chair = 5). Scores obtained via the MRC scale are presented in Table 1. Her mutation was the following: C.2771delA involving a 15-amino acid prolongation of the transportin-3 protein with dominant character.

This female was born prematurely and was artificially fed for maternal hypo-galactorrhea. Her psychomotor development was normal. Initially, she had difficulty running or climbing stairs and had an atrophy of the proximal lower limb muscles. Specific clinical indicators of the muscle impairment were the presence of skeletal abnormalities, such as arachnodactyly, pes cavus, and mild retractions of the toes and Achilles tendon. The muscle weakness slowly worsened over the years with a greater involvement of the proximal muscles of the upper limbs. As a child, this patient had some difficulty raising or lifting weights with arms at 5 years, and then she started presenting a lower limb involvement and was biopsied at 9 years. She has been followed for years by various centers since her LGMD diagnosis was missing, and at age 27 years, a spirometry showed an FVC 50% capacity. After walking for 150 m, she was fatigued. To rise from the chair, she used her arms bent on a table and walked up stairs using the rail. At 28 years, she was reexamined. On a neurological exam, she could walk with a slow waddling gait and was able to stand on tiptoes but not on heels. A second biopsy was performed and the transportin-3 defect was identified by next-generation sequencing. In the follow-up, her weakness increased and she now uses a wheelchair at 39 years of age.

### 3.1.4. Second Case of Family 2 (Patient 4)

This case centers on a 66-year-old woman of Spanish origin, weighs 59 kg, and her height is 159 cm. Grip strength is 9 kg and the Dexa scan was normal. She obtained a GSGC score of 21 (G = 4, S = 5, G = 7, C = 5). In conclusion, the patient shows a less severe phenotype with proximal weakness. Her mutation was the following: C.2771delA involving a 15-amino acid prolongation of the transportin-3 protein with dominant character.

The patient had difficulties when running at 6 years compared to her peers. During her adolescence, she was underweight, but then acquired weight, especially after her pregnancy. She went into menopause at 54 years old and is unable to rise from a chair or climb the stairs without clinging to the railing with both hands. A muscle MRI was performed and revealed a fatty substitution of posterior thighs. A CT head scan shows a hypodensity of white matter due to vascular chronic ischemia.

### 3.1.5. Third Case of Italo-Spanish Family (Patient 5)

A 33-year-old man of Spanish origin; his data on weight and height were not available in this case. He obtained a GSGC score of 18 (G = 5, S = 3, G = 7, C = 3). The LGMD was caused by a mutation of C.2771delA involving a 15-amino acid prolongation of the transportin-3 protein with dominant character. He had no cardiac involvement or cognitive impairment.

### 3.1.6. Fourth Case of Italo-Spanish Family (Patient 6)

A 38-year-old man of Spanish origin; similarly to the previous case, his data on weight and height were not available. He obtained a GSGC score of 27 (G = 7, S = 7, G = 7, C = 7). The LGMD was caused by a mutation of C.2771delA involving a 15-amino acid prolongation of transportin-3 protein with dominant character. He had no cardiac involvement or cognitive impairment.

### *3.2. Analysis of the Results*

Sociodemographic, GSGC, and INQoL data are listed in Table 2. Total INQoL score was 34. The worst results were in muscle weakness, fatigue, activities, relationships, and independence. On the other hand, the best scores were in muscle pain and emotions. Descriptively, there are not many differences between the scores of patients with different variants of TNPO3 mutations.

The analysis of sociodemographic variables, such as gender and age, and the subscales of the questionnaire indicated that there were no significant differences in terms of gender and the scores obtained on the different scales. On the other hand, there were differences between age and two subscales of QoL: social relationships and muscle pain. The older the

age, the greater the muscle pain (*rho* = 0.900, *p* = 0.037) and the greater the impact on their social relationships (*rho* = 0.900, *p* = 0.037).

**Table 2.** Sociodemographic characteristics and responses on the domains of INQoL in LGMD-D2. Scores obtained for each subscale are presented in percentage, and the GSGC scores represent the sum of the 4 parameters (gait, scale, Gowers, chair).

| Patient | Mutation | Age | Sex | GSGC | Muscle Weakness % | Locking % | Muscle Pain % | Fatigue % | Activities % | Independence % | Relationships % | Emotions % | Body Image % | INQoL Total Score % |
|---|---|---|---|---|---|---|---|---|---|---|---|---|---|---|
| *Patient 1* | c.2767delC | 17 | Male | 27 | 47.3 | 0 | 0 | 42.1 | 43.3 | 27.7 | 0 | 13.8 | 22.2 | 21.4 |
| *Patient 2* | c.2767delC | 48 | Female | 22 | 52.6 | 68.4 | 36.8 | 47.3 | 26.6 | 5.5 | 16.6 | 8.3 | 22.2 | 15.8 |
| *Patient 3* | C.2771delA | 39 | Female | 24 | 68.4 | 0 | 47.3 | 78.9 | 76.6 | 83.3 | 28.3 | 36.1 | 38.8 | 52.6 |
| *Patient 4* | C.2771delA | 66 | Female | 21 | 73.6 | 0 | 0 | 68.4 | 73.3 | 83.3 | 45.0 | 52.7 | 66.6 | 64.2 |
| *Patient 5* | C.2771delA | 33 | Male | 18 | 42.1 | 36.8 | 10.5 | 52.6 | 30.0 | 22.2 | 10.0 | 22.2 | 22.2 | 21.3 |
| *Patient 6* | C.2771delA | 38 | Male | 27 | 57.8 | 42.1 | 21.0 | 31.5 | 33.3 | 55.5 | 15.0 | 13.8 | 38.8 | 31.3 |
| *Total (M ± SD)* | | 40.1 ± 16.2 | | 23.1 ± 3.5 | 57.0 ± 12.1 | 24.5 ± 28.9 | 19.2 ± 19.5 | 53.5 ± 17.4 | 47.2 ± 22.2 | 46.2 ± 32.8 | 19.1 ± 15.6 | 24.5 ± 16.8 | 35.1 ± 17.4 | 34.4 ± 19.5 |

Additionally, a greater muscle weakness was correlated with a greater impact of the disease on social relationships (*rho* = 0.886, *p* = 0.019), on their independence (*rho* = 0.812, *p* = 0.050), and on their body image (*rho* = 0.926, *p* = 0.008).

Regarding the analysis of the responses reported by patients and caregivers, only three patients and three caregivers could be analyzed in the present study. This is because only three caregivers of the three patients who participated in the study wanted to take part in the study. Because of this, statistical analyses were not performed, as statistically significant conclusions cannot be extracted from such small groups. However, on a descriptive level, it can be seen in Table 3 that the means in each subscale are different, noting that caregivers reported higher scores in almost all spheres of impact on QoL. This indicates that, in general, they see the patient as having a worse QoL than the patient himself/herself thinks.

**Table 3.** Scores of INQoL responses of patients and caregivers obtained for each subscale are presented in percentage.

| Clinical Variables | Sociodemographic Variables | N | Total (M ± SD) |
|---|---|---|---|
| Muscle weakness | Caregiver | 3 | 72 ± 3 |
| | Patient | 3 | 56 ± 11 |
| Locking | Caregiver | 3 | 44 ± 39 |
| | Patient | 3 | 23 ± 39 |
| Muscle pain | Caregiver | 3 | 23 ± 39 |
| | Patient | 3 | 28 ± 25 |
| Fatigue | Caregiver | 3 | 75 ± 6 |
| | Patient | 3 | 56 ± 20 |
| Activities | Caregiver | 3 | 65 ± 19 |
| | Patient | 3 | 49 ± 25 |
| Independence | Caregiver | 3 | 76 ± 13 |
| | Patient | 3 | 39 ± 40 |
| Relationships | Caregiver | 3 | 39 ± 18 |
| | Patient | 3 | 15 ± 14 |
| Emotions | Caregiver | 3 | 46 ± 7 |
| | Patient | 3 | 19 ± 15 |
| Body image | Caregiver | 3 | 46 ± 19 |
| | Patient | 3 | 28 ± 10 |
| INQoL score | Caregiver | 3 | 56 ± 11 |
| | Patient | 3 | 30 ± 20 |

On the other hand, we analyzed whether there were differences between this dominant LGMD and a group of three people suffering from a recessive LGMD (Table 4). As discussed in the previous group, statistical analyses were not performed, as statistically significant conclusions cannot be extracted from such small groups. However, on a descriptive level, it can be seen in that the means in muscle weakness subscale and in independence subscale are particularly different. Also, a higher dependence produced by the recessive LGMD

is observed in the GSGC scale scores (Tables 2 and 4). In addition, in almost all areas of the INQoL scale, a greater impairment and a greater number of symptoms are seen in the recessive forms (Tables 2 and 4).

**Table 4.** Sociodemographic characteristics and responses on the domains of INQoL in recessive forms of LGMD. Scores obtained for each subscale are presented in percentage, and the GSGC scores represent the sum of the 4 parameters (gait, scale, Gowers, chair).

| Patient | Recessive Form | Age | Sex | GSGC | Muscle Weakness % | Locking % | Muscle Pain % | Fatigue % | Activities % | Independence % | Relationships % | Emotions % | Body Image % | INQoL Total Score% |
|---|---|---|---|---|---|---|---|---|---|---|---|---|---|---|---|
| *Patient 1* | *LGMD-R2* | 46 | Male | 27 | 57.8 | 57.8 | 0 | 63.1 | 40 | 27.7 | 5 | 11.1 | 33.3 | 23.4 |
| *Patient 2* | *LGMD-R5* | 47 | Female | 27 | 94.7 | 0 | 36.8 | 68.4 | 66.6 | 88.8 | 33.3 | 38.8 | 33.3 | 52.2 |
| *Patient 3* | *LGMD-R4* | 42 | Female | 27 | 84.2 | 73.6 | 0 | 68.4 | 53.3 | 88.8 | 8.3 | 19.4 | 44.4 | 42.8 |
| *Total (M ± SD)* | | 45 ± 2.6 | | 27 ± 0 | 78.9 ± 18.2 | 43.8 ± 38.8 | 12.2 ± 21.2 | 66.6 ± 3.0 | 53.3 ± 13.3 | 68.5 ± 35.2 | 15.5 ± 15.4 | 69.4 ± 14.2 | 37.0 ± 6.4 | 53.5 ± 14.6 |

## 4. Discussion

Transportinopathy is a dominantly transmitted LGMD entity due to the involvement of transportin-3, which results in a muscular dystrophy due to a mutation of this nuclear-import protein, which is also known to be involved in HIV virus transport. This pathology presents a heterogeneous clinical manifestation, described for the first occasion in a large Italo-Spanish family [12]. It is mainly characterized by two distinct clinical phenotypes: a severe muscle weakness of childhood onset impacting the pelvic girdle or a late-onset condition affecting both girdles. Both early and late-onset cases carry the microdeletion mutation in the TNPO3 gene and a single genotype can have different phenotypes that exhibit clinical histopathological, and MRI clinical features [31].

Previous studies have examined how LGMD affects people's QoL. Few studies across multiple subtypes have attempted to identify issues that contribute to disease burden [31]. However, none has examined LGMD-D2 subtype so far. When the disease is neither understood nor comprehended from the patient's perspective, the overall understanding of the disease may be scarce. Of particular relevance are patients' reports of their symptoms, as they allow for tailored and effective therapeutic intervention [32]. Our findings have captured the broad range of symptoms that most influence the lives of individuals suffering from LGMD-D2. The most affected were the independence, activities, and locking subscores, while the less affected were found for muscle pain and relationships.

On the other hand, it has been reported in the literature that the recessive forms have a greater progression and are more "severe". In the present study, muscle weakness and independence resulted in a greater impact on QoL in patients with a recessive form of LGMD. This is consistent with the fact that recessive forms may produce a higher dependence because of associated symptomatology [7,8]. However, these results should be interpreted with caution, as the groups were not homogeneous in terms of the number of participants and may therefore be biased. Moreover, there are no comparative studies of QoL between these two pathologies, and it would be interesting if such studies could analyze the differences in the impact of the disease between the various forms.

When our LGMD-D2 results are compared with another study that utilized INQoL in other recessive LGMD subtypes [17], it indicates that LGMD-D2 patients are more impaired in daily living activities, express higher levels of fatigue, and suffer more for locking consequences. In the case of studies where QoL has been assessed with other instruments in patients with recessive and dominant (non-transportinopathy) LGMD, our study and one conducted by Kovalchick et al. [18] report mobility problems and the impact on activities of daily living. In addition, fatigue and impact on independence and social relationships are also reported in the present sample [18].

When comparing these results with those of one study that was conducted with NMD and assessed with INQoL, the patients with LGMD-D2 are found to have higher scores on the impact of muscle pain and fatigue [33]. In another study in which patients with NMD were assessed using INQoL, it was observed that in the present sample, they had higher fatigue impact scores [34]. All aspects of QoL have a strong association with disability due

to NMD. According to the study by Bos et al. [35], although the deterioration of muscle function is the most severe limitation, an impaired mental function and pain are the most important predictive factors of QoL, followed by problems in carrying out tasks of daily living. Similarly, another aspect that influences QoL is fatigue. It is observed that more than 60% of people affected with NMD suffer from disabling fatigue [36].

Another important aspect recognized in the present study is that age seems to correlate with a greater involvement of the disease in activities of daily living and emotional management. Age as an influential variable in the worsening of disease symptoms is seen in other chronic pathologies [33]. The underlying explanation for this relationship seems to be that the specific pathophysiological mechanisms that cause clinical disorders are modified by an aging process [34]. However, due to the limited number of cases that could be included in the present study, it is difficult to make statements about how age worsens QoL, because two of the patients were 66 and 17 years old, so the results may be biased. Furthermore, one of the most important symptoms recognized by patients as having a major impact on their QoL was muscle weakness, affecting several domains such as social relationships, independence, and body image.

On the other hand, caregivers' reports are of particular relevance for a broader comprehension of the impact of the disease, as caregivers live with it, and often perceive the situation more objectively. Descriptively, when we analyzed the differences between patients' and caregivers' reports, it was found that caregivers reported a greater impact of the disease on their QoL in all domains except muscle pain. Regarding the differences between patients' and caregivers' reports, the differences found in almost all of the domains may be due to a lack of awareness of the limitations caused by the disease. The presence of unawareness of the disease has been reported in some studies conducted on people with NMD. In many cases, the diagnosis of a chronic illness can lead to defense mechanisms to reduce the stress produced by the stressful event. This results in a failure to objectively see the impact of the stressful event [37]. The objective report of symptoms and the impact on daily life to health professionals is of particular relevance. Patient care can be delayed by a number of factors, including the difficulty in identifying it for the first time, and this is a problem that affects new therapies, because they place special emphasis on early intervention. It is particularly important that the patient is aware of their symptomatology and how it affects their daily life. This becomes essential in rehabilitation, as without a full awareness of the disease, it may not be effective [38,39]. In conclusion, we have seen that this disease seems to have a higher impact on activities of daily living and independence, as well as higher locked symptoms due to contractions. Also, age and a greater muscle weakness may be important symptoms to take into account from the clinical perspective, as they may be an indicator of a greater impairment in other areas of their lives.

NMD, and especially LGMD, generate a series of symptoms that lead to a greater dependence and disability, which will ultimately result in a greater dependence on their environment, technical aids, and external people. Although this dependence in advanced stages of the disease becomes widespread, there are certain signs that will lead to a greater disability than others. Knowing the signs and how they affect lives helps professionals to direct the treatment in an optimal and effective way for the patient. Also, in the current context of ongoing research aimed at creating new treatments, it is particularly relevant to assess the impact on their QoL to know whether the treatment is really reducing their suffering.

The limitations of the present study refer to the difficulty in making comparisons between caregivers and patients, as not all caregivers of the affected persons wanted to participate. Another limitation is the small sample size, caused by the difficulty of recruiting patients with this pathology. This small sample limits comparisons between dominant and recessive forms, with only three patients with recessive forms, limiting statistical analyses. The patients presented have quite advanced functional states, so the data collected are probably influenced by the situation of these patients and their level of disability. In

addition, as discussed in the manuscript, there were two patients of widely different ages that could be influencing the results, limiting the conclusions that can be reached.

Future lines of research could comparatively analyze, among other neuromuscular pathologies, the impact that these diseases have, and if there are differences between them. An important aspect would be to understand whether dominant forms have a greater or lower impact on quality of life than recessive forms through research with a larger sample size. A larger sample of patients and caregivers would be useful in order to perform statistical analyses to obtain results about the differences in the reports. On the other hand, generating a protocol tailored to this disease that includes more evidence of the disease's impact would be ideal to analyze the effect of treatments on these patients.

Practical implications of the present study involve the knowledge of patients' self-reported symptoms, and the aspects of QoL with the greatest impact can be used as an outcome in the development of new therapies and the enhancement of clinical care. A better comprehension of the lack of awareness of the disease could improve prognosis, enable timely interventions, and support caregivers in early attention [40].

## 5. Conclusions

The present study is the first research that analyzes the QoL of people with LGMD-D2. Those affected have multiple symptoms that lead to disability and dependence, which ultimately lead to dependence on their environment. In the current context of constantly developing research for new treatments, it is essential to analyze which aspects are most affected. The present study shows that LGMD-D2 seem to have a greater impact on the activities of daily living, independence, the emotional sphere, and body image perception than other LGMD pathologies or other NMDs, considering the limitations of the comparisons, which were only at a descriptive level. It also appears that the impact is greater if the affected person is older, and that muscle weakness is a very disabling symptom in this variant. Finally, caregivers can be a very important tool in symptom reporting, as certain psychological adjustment mechanisms in the patient may be interfering with the objectivity of the report.

**Author Contributions:** Conceptualization, C.A. and A.A.R.; resources, C.A.; writing—original draft preparation, C.A., I.G., I.A. and A.A.R.; writing—review and editing, C.A., I.A. and A.A.R.; supervision, C.A., I.G. and A.A.R. All authors have read and agreed to the published version of the manuscript.

**Funding:** This work was supported by a Grant of the Education Department of the Basque Government (BOPV, 27th Juny 2019) (PRE_2019_1_0044).

**Institutional Review Board Statement:** This project has been approved by the Ethics Committee of the University of Deusto (Ref: ETK-39/18-19) and was conducted in accordance with the Declaration of Helsinki.

**Informed Consent Statement:** Participants were provided with an informed consent form. They had to sign it if they agreed to participate. In this consent form, they were informed that participation was voluntary and that they could leave at any time. They were also informed of the confidentiality of the data and that only the researcher would have access to them.

**Data Availability Statement:** The datasets generated and/or analyzed during the present study are not publicly available, as they belong to the University of Padua, but can be obtained from the corresponding author (Corrado Angelini) upon reasonable request.

**Acknowledgments:** We thank the patients and their families for their intensive cooperation in the study; we thank "Conquistando escalones" Italy and Spain for helping us to recruit families to participate in this study.

**Conflicts of Interest:** The author declares no conflict of interest.

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
