# Peer review of "Limb–Girdle Muscular Dystrophy D2 TNPO3-Related: A Quality of Life Study"

_muscles, doi:10.3390/muscles2030021_

Round 1
Reviewer 1 Report (Previous Reviewer 1)
The authors have addressed all my comments
Thank you
Author Response
We thank the reviewer for his insightful comments.
Reviewer 2 Report (Previous Reviewer 2)
The Authors addressed the reviewer's requests. The paper could be publishable in this form. However some typos must be correct (i.e. lines 122 They camefrom; line 192 weights28kg; the text font of lines 242-261)
Author Response
We thank the reviewer for his insightful comments and we have corrected the typos.
Reviewer 3 Report (Previous Reviewer 3)
Dear authors,
Although you have improved some parts, several important issues remain:
Introduction (less important): The introduction needs reduction. The newer section is even longer than the one before. I believe it can be reduced by 50%.
Material and methods (critical issue): authors derived the results based on 3 subjects in two of three groups. This is not enough! I will keep requiring at least 5-6 subjects in each group. Six is better. In other words, I do not mind supporting the manuscript, but the authors must go back to the patients and caregivers and find more probands. Regarding statistics: Is it possible to assess normality in groups of 3?
Results (also important): the section seems like a mixture of case reports with final group comparisons. What do authors what to present? I would place individual cases into supplementary material for readers interested in individual cases. However, the results should be only about group results.
I hope my notes can help improve the manuscript.
Author Response
We thank the reviewer for his insightful comments and we tried to reduce the introduction.
- Introduction (less important): The introduction needs reduction. The newer section is even longer than the one before. I believe it can be reduced by 50%.
The introduction has been reduced. We have tried to reduce as much as possible, as there are certain issues that could not be deleted because other reviewers have suggested them (For example, explanation of transportinopathies and their mutations).
- Material and methods (critical issue): authors derived the results based on 3 subjects in two of three groups. This is not enough! I will keep requiring at least 5-6 subjects in each group. Six is better. In other words, I do not mind supporting the manuscript, but the authors must go back to the patients and caregivers and find more probands. Regarding statistics: Is it possible to assess normality in groups of 3?
The comparison between groups has been eliminated, also in the case of dominant and recessive forms. Therefore, we only refer to mean differences at a descriptive level. Furthermore, in the limitations of the study and in future lines of research, the expansion of the sample is highlighted.
- Results (also important): the section seems like a mixture of case reports with final group comparisons. What do authors what to present? I would place individual cases into supplementary material for readers interested in individual cases. However, the results should be only about group results.
The usefulness of the paragraph of cases study is to show the progression of the disease from a clinical perspective. It is important to show the heterogeneity of clinical manifestations, even in situations where the same mutation is present. Therefore, we believe that it should be present in the results, as they are the outcomes of the medical evaluation.
Round 2
Reviewer 3 Report (Previous Reviewer 3)
Dear authors,
I am sorry for my previous review. It was quite difficult to navigate through the text, which also contained old changes. The current version is much better.
All my comments are in the uploaded pdf file.
I hope my comments would help you to improve your manuscript.
Best regards

Author Response
We thank the reviewer for his insightful comments.
First of all, we would like to apologize for the misunderstanding regarding the previous version of the manuscript. Due to the publisher's requirement to use "change control" for revisions, the document may be difficult to read. Thanks to your considerations, we have tried to improve the text in line with the proposed changes. As there are so many of them, they are not specified one by one, but they can be seen in the document through " track changes".
On the other hand, we would like to address in this letter some general points:
- Firstly, the different widths in the results section, there are impossible for us to change ourselves, we don't know why. We believe that it is an editing issue and that the publisher will be able to solve this problem.
- As to whether we can present higher resolution MRI images, we do not have higher resolution images; the machine was a 1.5 Thesla.
- Figure 2 and Figure 3 have been removed to avoid repeating information.
- On the other hand, as far as Table 3 is concerned, it is not possible to rotate it due to the formatting rules of the journal.
- Regarding the statistical analyses, and your proposal to perform a spearman's rank, these have been performed and no statistically significant differences have been found.
- With regard to the discussion and the section on anosognosia, the term anosognosia has been removed and explained in a more appropriate way to avoid misunderstanding.
- In the references section, all references have been thoroughly checked and put into Vancouver standards.
Thank you for your suggestions and we hope that the changes made have addressed your revisions.
This manuscript is a resubmission of an earlier submission. The following is a list of the peer review reports and author responses from that submission.
Round 1
Reviewer 1 Report
The study reports interesting findings on the quality of life in patients affected by LGMD, combining symptom reporting and clinical features. Including reports from caregivers is a plus of the study, but at present the authors do not highlight such novelty in full. Although reporting a limited number of cases, I do not see major issues in the data, however presentation needs improvement, and the final message could be delivered more efficiently.
Here my suggestions:
Major:
1-Title states TNPO, but there is no particular mention of the gene mutation (or specific comparison with other LGMDs) throughout the manuscript. While I understand that all analysed patients bear mutations in TNPO (do they?), it would be good to include a schematic of specific mutations (relevant to each patient) along TNPO sequence, and the effect on protein function (if known). It would be great to show whether a particular mutation associates with some QoL measurement better than others (eg higher/lower values).
Otherwise, while mentioned in introduction, TNPO should be removed from text and only cited once to disclaim that all patients present mutation in same gene.
2-Manuscript requires restructuring to enhance clarity and relevant points.
· the four detailed cases need to be reordered so that they are presented as numbered. Example, Patient 1: first presented and then the others. It is unclear why first case presented is patient 3 whereas the third case is patient 1. Reorder according to patient #number or renumber patients.
· 3- Each clinical case is described differently from another. Please follow the same order in all the four so that reader can easily compare if needed. Such as Age, Weight, Height, Grip Strength, GSGC and so on…
· 4-Where possible MRI images should be shown. Unless presented elsewhere, in this case it should be reported where.
· 5-Same applies to MRC score. Not all cases have MRC score, but where available it should be presented in a table. Having ‘Deltoid 3+/3+, Biceps 4-/ 3+, Triceps 3/ 3, Wrist flexors 4-/ 4-, Wrist extensor 4/ 4-, Thumb Abductor 3/ 3, Ilio-psoas 3+/ 3-,’ breaks the reading and is confusing.
· 6-Figure 2 should be improved. As it is presented now it fails to deliver the message. Indeed, it reports various different features of QoL, but it is unclear why author need/want to present all together and connected by lines. Instead separated graphs, one per features, reporting values from every patients, would best deliver the message, and show the differences. That would also make it less redundant with Table 1.
· 7-How are the rho values (and associated p values) related to TNPO mutation? Is there a particular mutation that may affect QoL in a more severe manner?
· 8-Why is INQoL presented as total across patients? It is difficult to understand the meaning of such data. However, it could be useful if presented in comparison to other LGMD: how does QoL in LGMD-D2 correlate with QoLs in calpainopathy and dysferlinopathy?
· 9-It is unclear to me the presence of INQoL in recessive forms of LGMD. Are these TNPO mutations? If not, what mutations? Are authors comparing INQoL values?
· 10-Table 2 is crucial to deliver the message stated in the abstract, but I think that it fails to do so. Why is it so important to have reports from caregivers? Please explain and maybe plot the results? It is usually easier for redears to understand graphs instead of being lost in tables (see ‘minor’ comment on U,P, R)
Minor:
· 1-Text should be checked for grammar and typos. Could be more concise and link between paragraphs could be strenghtened
· 2-Table 1 needs reformatting so that is easily readable and within page margins. As Muscles requires to work on a specific format, this should have been done prior to submission as what is seen is what is going to be published.
· 3-lease ensure that all References are formatted
· 4-he manuscript could be polished by removing several repeated concepts, especially in introduction. Eg on QoL between lines 64-78.
· 5-While the manuscript should be accessible to broad scientific audience, I suggest avoiding using terms such as ‘slim’. Line 213 ‘during her adolescence was slim’. Does not read/sound very clinical.
· 6-Similarly, line 219, what is a Dexa scan? Please reduce jargon to enhance accessibility
· 7-What are ‘U’, ‘P’ and ‘R’ in table 3?
Please double check for typos and readability in general, I think the flow could be enhanced.
Reviewer 2 Report
In this Manuscript Rodriguez and Collaborators analyzed the QoL of people with LGMD-D2. They found that LGMD D2 has a greater impact on activities of daily living, emotional space and body image perception. My major concern is that the proposed idea, i.e. the fact that LGMD D2 patients have a low quality of life compared to unaffected people has been already reported by others. Moroever the Authors do not distinguish the results obtained in young adults (17 years old patients) from those obtained in 33-48 years old and most importantly from a 66-years old patient. Indeed ageing per se may influence some of the analyzed parameters. For all of these reasons the Manuscript could not be published in muscles.
Other points:
1) It is not clear form table 1 whether all the values are indicated in %
2) Figure 1 and Figure 2 report the very same results.
3) Figure legends are always lacking.
Morover some analyzed parameters like "emotions", "Body image" are not well characterized.
Reviewer 3 Report
See the attached file
